| Editor's Pick | Systems Biology | Research Article

# Integrated multi-omics reveals coordinated *Staphylococcus aureus* metabolic, iron transport, and stress responses to human serum

Warasinee Mujchariyakul,[1] Calum J. Walsh,[1,2] Stefano Giulieri,[1,2,3] Cameron Cramond,[1] Kim-Anh LêCao,[4] Timothy P. Stinear,[1,2] Benjamin P. Howden,[1,2,5] Romain Guérillot,[1,2] Abderrahman Hachani[1,2]

**ABSTRACT** Bloodstream infections caused by *Staphylococcus aureus* remain a leading cause of mortality worldwide. Our understanding of *S. aureus* survival and persistence in human serum, a cell-free fraction of blood hostile for bacteria, is still limited. Here, we applied multivariate data integration methods and network analysis to a multi-omic data set generated from five clinically prevalent *S. aureus* genotypes exposed to human serum. We observed, and then confirmed using isogenic mutants the significant roles of *gapdhB*, *sucA*, *sirA*, *sstD*, and *perR* in bacterial survival in serum. These data show that metabolic versatility in carbon source usage, iron transport, and resistance to oxidative stress is interlinked and central to *S. aureus* fitness in serum, representing potential *S. aureus* vulnerabilities that could be exploited therapeutically.

**IMPORTANCE** Bloodstream infections caused by *Staphylococcus aureus* are associated with mortality rates of up to 30%. However, the molecular mechanisms that enable this pathogen to survive in human serum—a nutrient-limited and immunologically hostile environment—remain poorly understood. By integrating multi-omic data from five clinically relevant *S. aureus* genotypes and validating key signatures using mutants, we identified conserved genetic determinants critical for bacterial survival in serum. Our findings highlight the interconnected roles of carbohydrate metabolic flexibility, iron acquisition, and oxidative stress resistance in shaping *S. aureus* adaptation to serum. This work advances our understanding of microbial strategies to survive in the bloodstream and demonstrates the potential of multi-omic integration to uncover therapeutic vulnerabilities in bacterial pathogens.

**KEYWORDS** *Staphylococcus aureus*, multi-omics data integration, metabolism, iron acquisition, stress responses, bacteremia

$S$taphylococcus aureus is a significant opportunistic human pathogen (1). As a commensal, *S. aureus* coexists with other members of the human microbiome in up to 30% of the human population, typically colonizing the skin or nares, with the potential to become a lethal pathogen (2). *S. aureus* bloodstream infections occur at a rate greater than 10 per 100,000 people per year, with the severity of infections compounded by the widespread emergence of resistance to last-line antibiotics, such as vancomycin, and mortality rates reaching 30% (3).

During bloodstream infections, professional phagocytes and non-cellular immune components, including antimicrobial peptides (AMPs), immunoglobulins, and the complement system, target invasive pathogens for clearance (4, 5). While serum does not contain immune cells, it represents a potent bactericidal environment for blood-invasive bacterial pathogens (6–8)—an environment against which *S. aureus* has evolved multiple

**Peer Reviewer** Oscar Quijada Pich, Institut d'Investigacio i Innovacio Parc Tauli, Sabadell, Spain

Address correspondence to Abderrahman Hachani, abderrahman.hachani@unimelb.edu.au.

Benjamin P. Howden, Romain Guérillot, and Abderrahman Hachani contributed equally to this article.

The authors declare no conflict of interest.

See the funding table on p. 19.

defense mechanisms (1, 9, 10). The bacterium can evade humoral immunity by directly interfering with antibody functions to prevent opsonization for phagocytosis (11, 12) or by degrading complement components (13). *S. aureus* can inactivate AMPs through proteolytic degradation (14), modification of its surface charge, or remodeling both the organization of its cell wall peptidoglycan and the abundance of cardiolipin in its plasma membrane. The latter two processes also contribute to *S. aureus* resistance to cell wall-targeting antibiotics (15). The host can also impose nutritional immunity by sequestering essential nutrients, primarily trace metals, such as iron, zinc, and manganese, to restrict bacterial growth. Bacterial pathogens have evolved mechanisms to circumvent nutritional immunity (16). For example, *S. aureus* has multiple trace metal-sequestering strategies that also contribute to its survival in blood (17).

Elucidating the metabolic reprogramming of *S. aureus* in response to serum-derived stressors—such as nutrient limitation and acellular immune factors—can uncover fundamental metabolic pathways that facilitate its adaptation, survival, and persistence in the bloodstream. These insights may inform the development of targeted antimicrobial strategies and rationally designed interventions to mitigate invasive and chronic *S. aureus* infections.

We previously reported a comprehensive multi-omics analysis of four clinically significant sepsis-causing bacterial pathogens (*Escherichia coli*, *Klebsiella pneumoniae*, *S. aureus*, and *Streptococcus pyogenes*) following their exposure to human serum (18). This work revealed a set of conserved adaptive responses across these phylogenetically diverse species.

Notably, all four pathogens exhibited a coordinated upregulation of lipid metabolic pathways that was associated with extensive remodeling of the cell envelope. This adaptation likely reflects a shared osmoprotective strategy, enhancing bacterial survival in the hostile and nutrient-limited environment of human serum. Building on our multi-omics data set, this study examines the serum response of *Staphylococcus aureus* across five clinically relevant clonal lineages—BPH2760 (ST1), BPH2819 (ST5), BPH2900 (ST22), BPH2947 (ST239), and BPH2986 (ST8). The aim is to identify a conserved set of molecular features that drives species-specific adaptation to serum during septicemia.

With a focused analysis on *S. aureus*, we refined our analytical workflow here. In our previous study (18), we used a late integration approach, independently analyzing transcriptomic, proteomic, and metabolomic data sets before mapping significant features to metabolic pathways and Gene Ontology terms. While this revealed broad functional patterns, it lacked the resolution to detect coordinated molecular changes across omics layers or to uncover direct cross-omic relationships. To address these limitations, we adopted the DIABLO framework (Data Integration Analysis for Biomarker discovery using Latent cOmponents) from the mixOmics R package, which applies a sparse, multiblock partial least squares (PLS) method, optimized to integrate all omics data sets within a single multivariate model. By selecting a minimal set of features that co-vary across data types and distinguish between serum- and RPMI-exposed conditions, DIABLO enables the identification of cross-omic interactions, reduces the statistical burden of separate analyses, and preserves the integrated biological context of the *S. aureus* serum response (19, 20). We further interrogated these signatures using differential expression and network-based analyses and identified interconnected metabolic pathways involved in serum adaptation through overrepresentation analysis (ORA) and gene set enrichment analysis (GSEA).

Our integrated omic analysis shows that *S. aureus* maintains optimal fitness in serum by undergoing changes in key carbohydrate metabolic processes, iron acquisition, and resistance to oxidative stress pathways. This combined response likely enables *S. aureus* to withstand host stressors, such as nutritional immunity, and promotes survival in human serum.

## RESULTS

### Coordinated transcriptomic, proteomic, and metabolomic adaptation of *S. aureus* to serum

We performed multi-omics analysis to find the shared molecular responses to human serum of five *S. aureus* clinical isolates (BPH2760, BPH2819, BPH2900, BPH2947, and BPH2986) (Fig. 1A). Principal Component Analysis (PCA), a dimensionality reduction method, was applied to each omics layer, revealing clear separation between serum- and RPMI-exposed *S. aureus*. The first principal component (PC1) explained 24%, 22%, 27%, and 29% of the total variance in the transcriptomic, proteomic, and metabolomic (GC-MS and LC-MS) data sets, respectively, capturing the dominant condition-specific signal (Fig. 1B). We used DIABLO to integrate the data sets into a unified predictive model of *S. aureus* exposure to human serum. DIABLO is a supervised multiblock method based on sparse partial least squares discriminant analysis (sPLS-DA) (19, 20). DIABLO uses the PLS framework to model relationships between molecular features (derived from transcriptomic, proteomic, or metabolomic data sets) and phenotypic outcomes. By integrating multiple omics data sets (defined as multiblock) and applying feature selection (sparsity), DIABLO highlights the subset of variables with the greatest discriminatory power—in this case, the staphylococcal molecules most strongly differentiating growth in RPMI versus serum (Fig. 1C and D; Fig. S1A through C). The sample plot showed that the first component captured most of the common serum responses across all five *S. aureus* strains, while the second component reflected strain-specific variation (Fig. S1C). The final DIABLO model used 60 multi-omic features (see methods) that optimally discriminated serum from RPMI exposure on the first component (Table S1; Fig. S1D).

A coordinated serum response signature was indicated by highly correlated latent components across the omics layers, including transcriptomic, proteomic, and metabolomic. The signatures extracted from the omics layers were highly correlated (Pearson's *r* > 0.95) and indicated both class separation and balanced correlation between the data sets (Fig. 1C). Loading values from the DIABLO model (Fig. 1D) show the specific contribution of individual variables, with higher absolute values marking greater discriminative power.

To explore consistent patterns between omics data, a similarity matrix was calculated across all components and selected features (Table S1) and visualized as a network (Fig. 1E) (21). The network revealed interactions across transcriptomic, proteomic, and metabolomic layers, with features grouped by function and showing both positive and negative correlations, suggesting a coordinated response to serum. The staphylococcal pathways enriched among the discriminant and correlated features identified by DIABLO included cell wall biosynthesis, carbohydrate metabolism, iron transport, and defense against host immunity (Fig. 1D and E; Table 1). Genes and their products central to these pathways exhibited strong positive correlations: (i) *glmS*, pivotal in cell wall biosynthesis and lipid metabolism (via linoleic and oleic acids, and cholesterol metabolites); (ii) *gapdhB* and *sucA*, involved in carbohydrate metabolism, and *sstD*, *scdA*, and *isdB*, contributing to iron uptake; and (iii) *perR* and *sbi*, supporting oxidative stress resistance mechanisms and immune evasion, respectively. In contrast, genes involved in nucleotide metabolism (*mtaB*, *miaB*, *guaC*) were negatively correlated with carbohydrate metabolism genes (*gapdhB* and s*ucA*) and lipids (linoleic and oleic acids).

The observed patterns show that *S. aureus* undergoes substantial metabolic adaptation in response to serum, engaging in the uptake and utilization of serum-derived amino acids, carbohydrates, fatty acids, nucleotides, and their derivatives. Notably, the analysis revealed a marked increase in the abundance of staphylococcal iron acquisition proteins upon serum exposure. In the host environment, extracellular iron is limited and typically sequestered by high-affinity iron-binding proteins, such as transferrin, lactoferrin, and hemoglobin within erythrocytes. The elevated levels of both siderophore-dependent (HtsA, SirA, SstA) and siderophore-independent (IsdB) iron acquisition proteins underscore the critical role of iron sequestration in *S. aureus* survival (Fig. 1D). The integration of multi-omic data reveals the responses that support metabolic

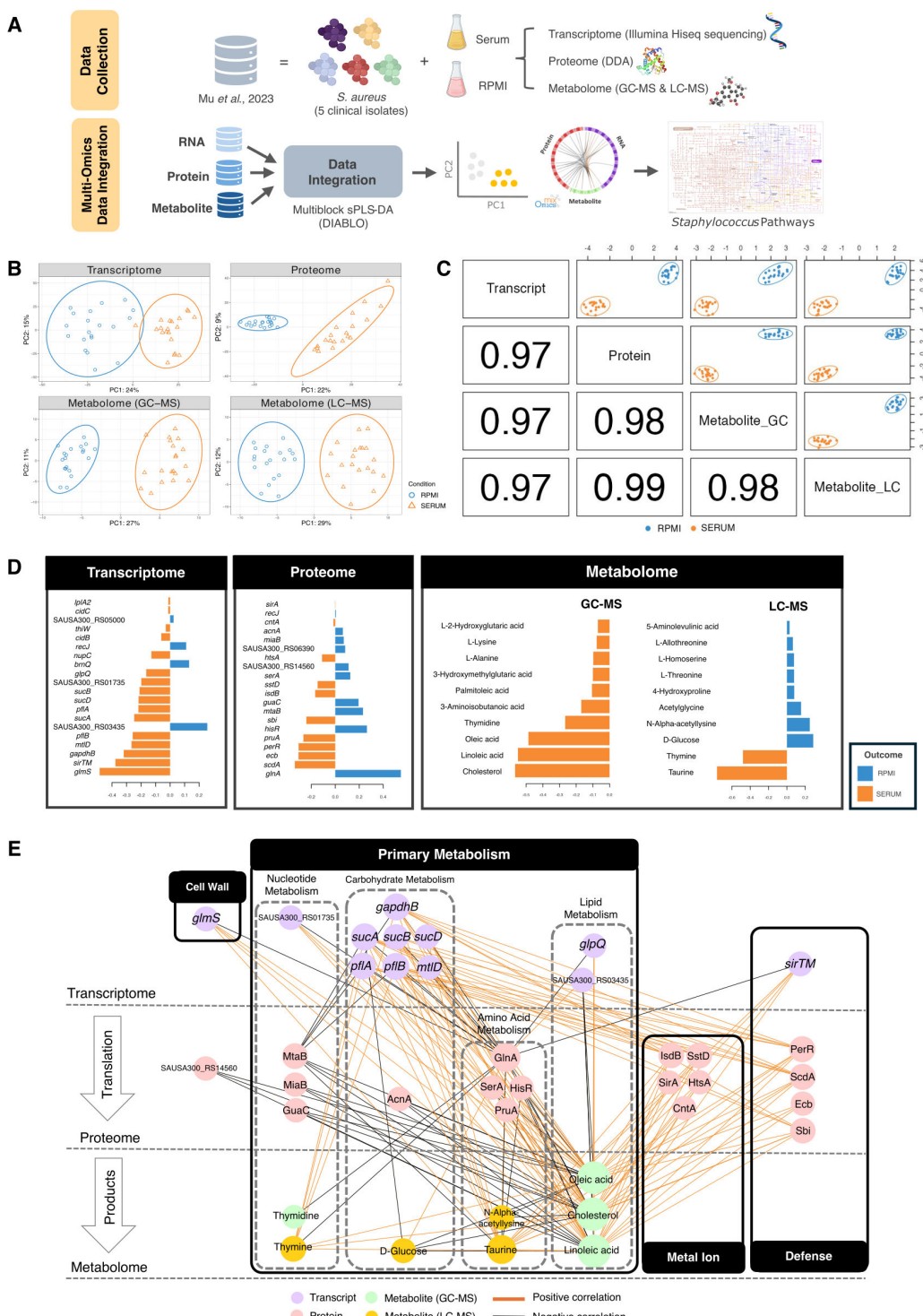

**FIG 1** Multi-omics identification of molecular signatures of *S. aureus* adaptation to serum. (A) Schematic workflow of multi-omics analysis to predict *S. aureus* signature molecules in response to human serum. (B) Unsupervised analysis using principal component analysis (PCA). Sample plot from PCA illustrating the consistency of sample clustering across data sets (transcriptomics, proteomics, and metabolomics [GC-MS and LC-MS]), with samples labeled based on medium conditions (SERUM and RPMI). (C) Supervised analysis using multiblock sPLS-DA. Diagnostic plot (sample scatter plot), according to medium conditions (SERUM and RPMI). The numbers in each block below the diagonal represent correlation coefficients between the first components of each data set. (D) Loading plot depicting features selected as optimally discriminatory between conditions by PLS-DA from the first component in each data set. (E) Correlation network visualizing pairwise

**Fig 1 (Continued)**

correlations (> |0.9|) between variables (transcripts, proteins, metabolites). Edge connections represent correlations, with edge colors indicating positive (orange) or negative (black) correlations. Node sizes are proportional to the number of interactions, and node colors represent data types: purple (transcripts), pink (proteins), green (GC-MS metabolites), and yellow (LC-MS metabolites).

resilience, enabling *S. aureus* to maintain fitness while countering intrinsic serum stressors.

## Network and enrichment analyses reveal *S. aureus* shared and strain-specific responses to serum

Differentially expressed (DE) genes or proteins for each strain were identified and further subjected to an intersection analysis. This analysis showed that at the transcript level, 143 genes were upregulated, and 65 genes were downregulated in response to serum. At the proteomic level, there were 21 upregulated and 17 downregulated proteins in response to serum (Fig. S2A and B; Table S2). We performed a pathway enrichment analysis (PEA), including an ORA and a GSEA, to reduce the complexity of data and find overrepresented pathways elicited upon exposure to serum. ORA is widely used to identify overrepresented biological functions in DE gene sets, performing best when handling wide gene expression differences. Genes were filtered using criteria of log2 fold change (log2FC) > 1 and adj-*P* value < 0.05. To improve the granularity of PEA outputs, a GSEA analysis was also applied to capture pathways where gene expression changes were more subtle but consistently enriched, enabling the detection of significant yet lower-magnitude responses. Using the similarity in gene subsets shared between pathways, we also conducted a network analysis to find interactions among the enriched pathways across the omics layers (Fig. 2A; Fig. S3A through J; Table S3). Consistent with previous DIABLO findings, ORA and GSEA identified four distinct functional modules comprising carbohydrate metabolism, ribosome-associated pathways, iron acquisition, and nucleotide metabolism (Fig. 2B through E).

The carbohydrate metabolism module was composed of the glycolysis/gluconeogenesis pathway, along with other carbohydrate-processing routes, such as phosphotransferase systems (PTS) (Fig. 2B and C). Upregulated carbohydrate transporter genes, such as *uhpT*, *mtlA*, *fruA*, *glcC*, *lacE*, SAUSA300_RS01760, and *rbsU*, support the usage of various saccharides, including glucose, fructose, lactose, D-mannitol, and ribose (Table S2). Once transported and phosphorylated by PTS, carbohydrates, such as glucose and fructose, are processed by glycolysis and converted into ribose-5-phosphate, glyceraldehyde-3-phosphate, and fructose-6-phosphate intermediates. These key metabolites bridge glycolysis to nucleotide and lipid metabolism (49, 50) (further detailed in Fig. 4), thereby enhancing the staphylococcal metabolic flexibility in serum. In contrast to other metabolic modules, our network analysis revealed that ribosome-associated pathways were largely isolated, exhibiting no direct connectivity with broader metabolic networks (Fig. 2C, Fig. S3A through D). Transcriptome profiling indicated a suppression of ribosome biogenesis following serum exposure, characterized by the downregulation of genes encoding 30S and 50S ribosomal subunit proteins (Tables S2 and S3). Expression of the translational GTPase *typA* (also known as *bipA*), an important regulator of ribosome assembly and translational fidelity under stress conditions (51, 52), was also significantly reduced (Table S2). These findings show that protein biogenesis is downregulated as part of a stress-adaptive response of *S. aureus* to serum.

Among the proteomic changes induced by serum exposure, iron acquisition and nucleotide metabolism pathways showed the most substantial alterations (Fig. 2D and E; Fig. S2B). Notably, there was a marked increase in the abundance of iron transport proteins, including SirA, FhuC, SstC, SstD, HtsA, IsdB, IsdE, and IsdI. In addition, elevated levels of ScdA and CntA (53)—proteins involved in iron-sulfur cluster binding—alongside the downregulation of *miaB* (54), suggest the activation of iron-sparing strategies by *S. aureus* under serum conditions (Tables S2 and S3). The significant increase in levels

**TABLE 1** Functions of selected variables shown in Fig. 1E

| Variable name | Function(s) | Reference(s) |
|---|---|---|
| *acnA/citB* (SAUSA300_RS06765) | TCA cycle | (22) |
| Alpha/beta hydrolase (SAUSA300_RS03435) | Putative lipase/esterase. Maintenance of bacterial homeostasis and mediation of host-pathogen interactions | (23, 24) |
| *cntA* (SAUSA300_RS13350) | Transport of multiple metals such as nickel, cobalt, zinc, iron, copper, and manganese | (25–27) |
| *ecb* (SAUSA300_RS05670) | Complement binding | (28) |
| *gapdhB* (SAUSA300_RS08910) | Gluconeogenesis | (29) |
| *glpQ* (SAUSA300_RS04655) | Utilization of glycerophosphodiesters (GroPC, GroPI, GroPS, GroPE, and GroPG) and release of glycerol-3-phosphate | (30) |
| *glnA* (SAUSA300_RS06485) | Synthesis of L-glutamine from L-glutamate | (31) |
| *guaC* (SAUSA300_RS06695) | Conversion of GMP to IMP | (32) |
| *hstA* (SAUSA300_RS11760) | Substrate-binding protein in siderophore-dependent iron acquisition pathways | (33) |
| Hypothetical protein/hisR (SAUSA300_RS10330) | Repressor in histidine biosynthesis | (24) |
| *isdB* (SAUSA300_RS05535) | Substrate-binding protein in siderophore-independent iron acquisition pathways (heme–iron uptake pathway), interaction with von Willebrand factor, promotion of adherence to endothelial cells | (34, 35) |
| *sbi* (SAUSA300_RS13060) | Complement binding | (36) |
| *scdA* (SAUSA300_RS01350) | Repair of oxidative and nitrosative damage to iron-sulfur centers and role in cell wall metabolism | (37, 38) |
| *serA* (SAUSA300_RS09115) | Biosynthesis of L-serine | (39) |
| *sirA* (SAUSA300_RS00605) | Substrate-binding protein in siderophore-dependent iron acquisition pathways | (33) |
| *sirTM* (SAUSA300_RS01740) | Regulation of oxidative stress | (40) |
| *sucA* (SAUSA300_RS07105) | TCA cycle | (22) |
| *sucB* (SAUSA300_RS07100) | TCA cycle | (22) |
| *sucD* (SAUSA300_RS06165) | TCA cycle | (22) |
| *sstD* (SAUSA300_RS03880) | Substrate-binding protein in siderophore-dependent iron acquisition pathways | (33) |
| *miaB* (SAUSA300_RS06405) | Modification of tRNA | (41) |
| *mtaB* (SAUSA300_RS08375) | Modification of tRNA | (41) |
| *mtlD* (SAUSA300_RS11620) | Conversion of mannitol-1-P to fructose-6-P | (42) |
| *perR* (SAUSA300_RS10060) | Sensing of hydrogen peroxide ($H_2O_2$) | (43) |
| *pflA* (SAUSA300_RS01160) | Role in formate metabolism | (44) |
| *pflB* (SAUSA300_RS01155) | Role in formate metabolism and promotion of biofilm persistence | (44, 45) |
| Polyisoprenoid-binding protein (SAUSA300_RS14560) | Role in the electron transport system | (32, 46) |
| Protein-ADP-ribose hydrolase (SAUSA300_RS01735) | Reversal of ADP-ribosylation in posttranslational modification | (47) |
| *pruA/rocA* (SAUSA300_RS13840) | Catabolism of proline to glutamate | (48) |

of ScdA and CntA, iron cluster-binding proteins, concomitant with the downregulation of *miaB*, points to the engagement of staphylococcal iron-sparing strategies during serum exposure (Tables S2 and S3). Thus, *S. aureus* coordinates enhanced iron uptake

with the regulation of iron incorporation into metalloproteins to efficiently manage iron availability during serum exposure (Fig. 2D; Fig. S3B and F).

Proteins involved in nucleotide metabolism, particularly those associated with the inosine monophosphate (IMP) biosynthetic pathway, were prominently increased across all *S. aureus* isolates (Fig. 2E; Fig. S3H). This pathway is central to the synthesis of purine nucleotides, such as adenosine monophosphate (AMP) and guanosine monophosphate (GMP), which are critical molecules for DNA replication, RNA transcription, and cellular proliferation. These shifts underscore the prioritization of iron homeostasis and nucleotide biosynthesis as key adaptive strategies employed by *S. aureus* to support survival and growth in serum.

## Experimental validation of predicted molecular signatures of *S. aureus* responses to human serum

To confirm the findings from the multi-omic integration and pathway enrichment analyses, we assessed the fitness of *S. aureus* transposon mutants in genes involved in carbon metabolism, iron acquisition, and oxidative stress resistance, relative to their parental wild-type USA300 JE2 strain, in the presence of serum (Fig. 1E and 3A). We initially verified the genetic relatedness of the five *S. aureus* bloodstream isolates to the laboratory strain JE2, which was used for this phenotypic validation. We observed a high conservation of the *S. aureus* core genome (2,222 core genes—representing 71% of JE2 genome—and 903 accessory genes). Strain JE2 was most closely related to the clinical isolate BPH2986, with both belonging to the ST8 USA300 lineage and differing by only 73 single nucleotide polymorphisms (SNPs) (Fig. S4A through D).

The DIABLO integrative model identified *gapdhB* and *sucA* as key contributors to response to serum, exhibiting the highest loading values (Fig. 1D; Table S1). Disruption of either gene resulted in a significant reduction in bacterial fitness in serum, showing they both play important roles in response to serum exposure (Fig. 3B; Fig. S5A).

GapdhB, a gluconeogenic enzyme, and SucA, an enzyme of the tricarboxylic acid (TCA) cycle, represent distinct nodes in the carbon metabolism pathway. This functional difference suggests that their contributions to *S. aureus* fitness may be contingent upon the availability and type of carbon substrates. To test this, we compared the growth and respiration phenotypes of *gapdhB* and *sucA* transposon mutants with 190 different carbon sources (Biolog Inc., Hayward, CA, USA) to those of the wild-type *S. aureus* JE2 strain. These carbon sources fell into several broad categories, including carbohydrates (mono-, di-, and oligosaccharides), carboxylic acids (short-chain fatty acids such as acetate, dicarboxylic acids such as fumarate, and hydroxy acids such as lactate), amino acids, alcohols, nucleosides, and other organic compounds. The metabolic activities of *S. aureus* JE2 wild type (WT), *gapdhB,* and *sucA* transposon mutants were more strongly influenced by individual carbon sources—thus compound-specific—than by their broader carbon classification (Fig. S6A).

Among the tested carbon sources, methyl pyruvate, α-D-glucose, D-glucosamine, and other hexoses significantly enhanced respiration in both WT and mutants, whereas glyoxylic acid and itaconic acid suppressed respiration (Fig. 3C; Fig. S6; Table S4). Notably, methyl pyruvate supplementation partially rescued the respiratory deficiency of the *sucA* mutant, surpassing WT levels, while it failed to restore *gapdhB* mutant activity to WT levels (Fig. 3C; Fig. S6B). While we showed that the metabolic responses of WT and mutant strains were shaped more by specific carbon compounds rather than by their general category, supplementation with methyl pyruvate revealed distinct roles for *gapdhB* and *sucA* in pyruvate catabolism (Fig. S6B). Methyl pyruvate, a membrane-permeable pyruvate analog, enters cells where it is converted to acetyl-CoA. In the absence of *sucA,* supplementation with methyl pyruvate provides an alternative carbon source that bypasses the glycolytic block caused by the absence of conversion of 2-oxaloglutarate to succinyl-CoA in the TCA cycle. GapdhB catalyzes the conversion of glyceraldehyde-3-phosphate (GA3P) to 1,3-bisphosphoglycerate (BPG) in the gluconeo-

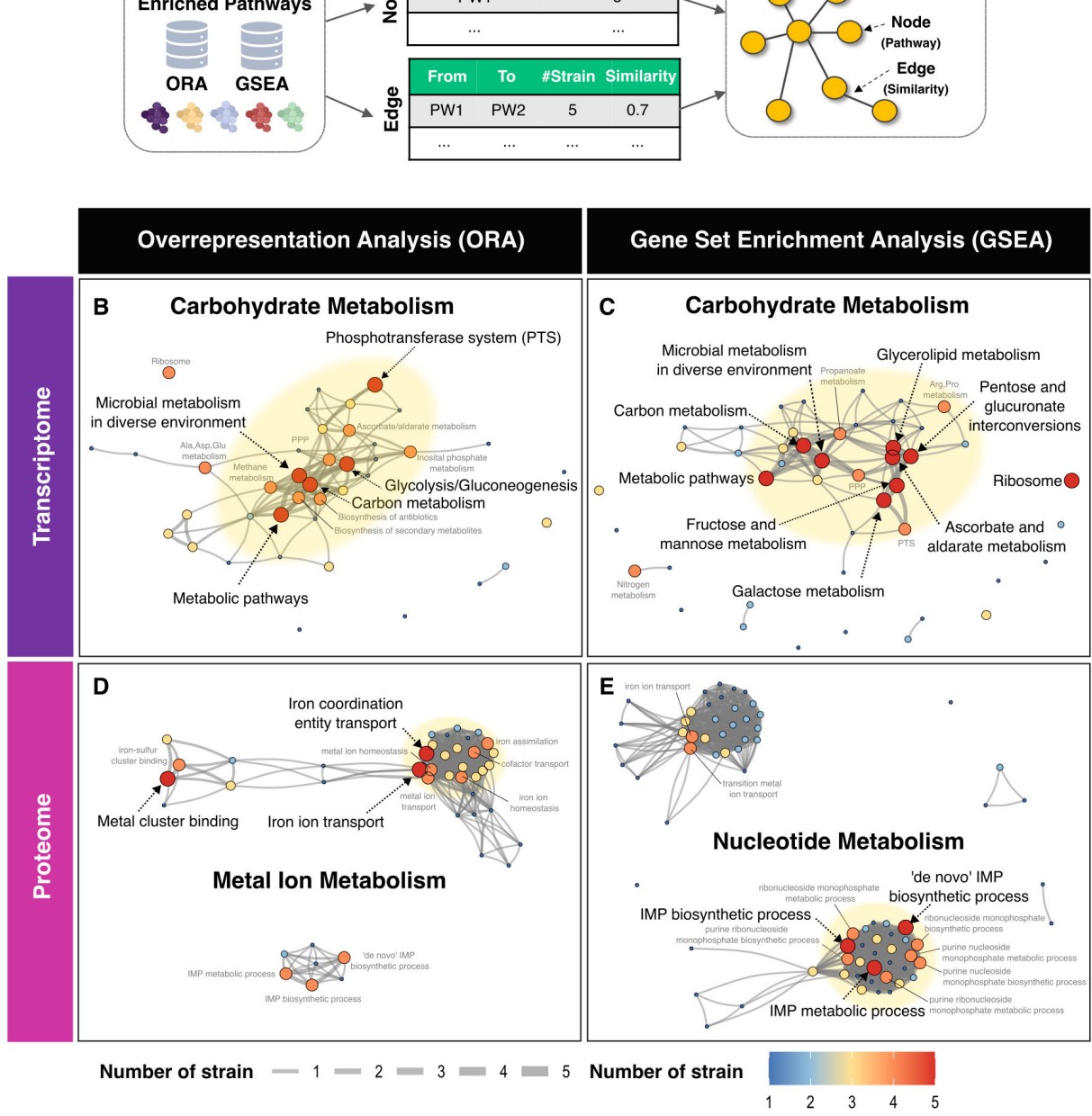

**FIG 2** Clustering of transcriptomic and proteomic data into function-related modules. (A) Workflow overview for network analysis of enriched pathways identified from overrepresentation analysis (ORA) and gene set enrichment analysis (GSEA). Nodes represent pathways, and edges represent pairwise similarity between enriched terms or pathways (PW), calculated using Jaccard's similarity coefficient. (B–C) ORA and GSEA of transcriptomic data. (D–E) ORA and GSEA of proteomic data. Dot sizes and colors represent the number of strains sharing the same enriched pathways. Edge widths indicate the number of strains with interactions between pathways, while edge lengths reflect pathway similarity. Yellow shading highlights denote functionally related modules.

genic direction (29). In a *gapdhB* mutant, the gluconeogenic pathway is blocked, so cells cannot convert methyl pyruvate (see primary metabolism in Fig. 4).

To investigate the contribution of iron acquisition systems to *Staphylococcus aureus* fitness in serum, we evaluated four mutants, with transposon insertions disrupting genes *htsA*, *sirA*, *sstD*, and *isdB*. The *htsA* and *sirA* genes encode substrate-binding proteins associated with the siderophores staphyloferrin A (SA) and staphyloferrin B (SB), respectively, both of which facilitate iron extraction from transferrin in serum (33). The gene *sstD* encodes a component of the Sst system, which mediates iron uptake from

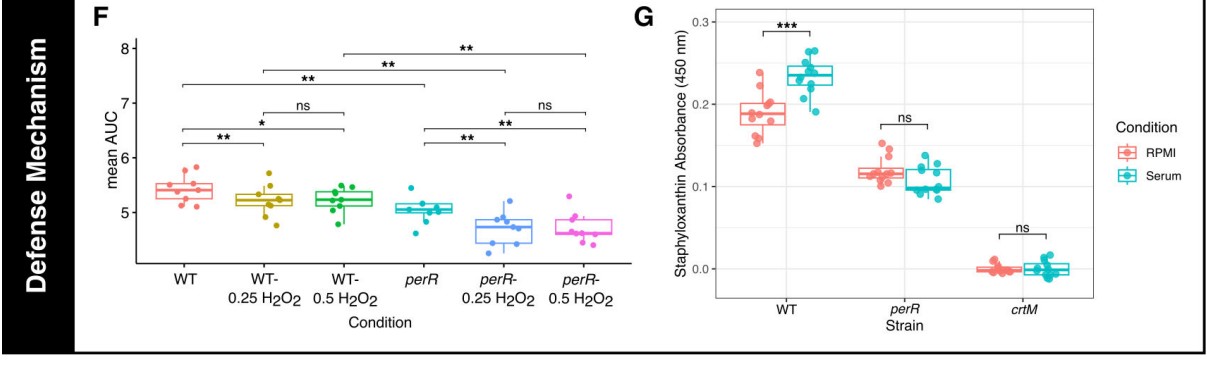

**FIG 3** Phenotypic validation of molecular signatures of serum adaptation and metabolic characterization. (A) Workflow of phenotypic validation experiments to confirm the predicted molecular signatures of *S. aureus* exposed to human serum. (B) Box plots comparing area under the curve (AUC) values of *gapdhB* and *sucA* mutants to wild-type *S. aureus* (WT-JE2) grown in the presence of serum for 20 h (*n* = 6 biological replicates). (C) Fitness profiling of *S. aureus* represented by the rate of bacterial respiration in the presence of different carbon sources (corresponding to metabolic activity inferred from the reduction of tetrazolium). Plots comparing area under the curve (AUC) values of *gapdhB* and *sucA* mutants to wild-type *S. aureus* (WT-JE2) grown in the presence of sole carbon sources for 24 h (*n* = 3 biological replicates). (D) Representation of iron acquisition pathways in *S. aureus*. (E) Box plots comparing AUC values of iron transport-deficient mutants (*htsA, sirA, sstD, isdB*) to WT-JE2 cultured in serum for 20 h (*n* = 6 biological replicates). (F) Box plots comparing AUC values of *perR* mutant to WT-JE2 grown in serum with different concentrations of $H_2O_2$ for 20 h (*n* = 9 biological replicates), demonstrating the role of *perR* in oxidative stress resistance. (G) Quantification

Fig 3 (Continued)

of staphyloxanthin production by WT-JE2, *perR,* and *crtM* mutants, following 5 h of incubation in RPMI or serum ($n$ = 12 replicates, generated from four different cultures using two serum batches). All data points represent mean values of technical replicates, for at least three independent biological replicates, with error bars indicating the corresponding standard deviations from the means. Statistical significance between WT-JE2 and isogenic mutants was determined using the Wilcoxon paired test. Significance levels are indicated by asterisks: *$P < 0.05$, **$P < 0.01$, and ***$P < 0.001$. ns: non-significant.

host-derived catechols (33). In contrast, *isdB* encodes a cell wall-anchored hemoglobin receptor within the Isd (iron-regulated surface determinant) system, a siderophore-independent pathway that enables heme iron acquisition (Fig. 3D). Proteomics revealed significantly elevated levels of HtsA, SirA, SstD, and IsdB following serum exposure, indicating their upregulation in response to iron-limited conditions (Fig. 1D; Fig. S2B). *S. aureus sirA* and *sstD* transposon mutants had significantly impaired fitness in serum, consistent with the importance of SB-mediated and catechol-mediated iron uptake under these conditions (Fig. 3E). In contrast, disruption of *htsA* did not significantly affect fitness, aligned with previous findings that SB biosynthesis, unlike SA, is independent of TCA cycle activity and may be preferentially utilized in iron-restricted environments (55). The reduced fitness observed in the *sirA* mutant likely reflects the impaired utilization of SB as an iron capture process in serum.

Interestingly, the *isdB* mutant exhibited no fitness defect in commercially sourced serum, likely due to the absence of hemoglobin or heme (56), which are typically sequestered within red blood cells and thus unlikely present in commercial cell-free serum preparations. These findings underscore the context-dependent utilization of iron acquisition systems in *S. aureus* and highlight the critical role of siderophore- and catechol-mediated pathways in supporting staphylococcal survival in iron-limited host environments.

Oxidative stress constitutes a fundamental component of the host immune response, primarily through the generation of reactive oxygen species (ROS) aimed at neutralizing invading pathogens. To counteract ROS, bacteria have evolved intricate oxidative stress response mechanisms (57, 58). Central to these systems are redox-sensitive transcriptional regulators, such as OhrR, OxyR, and PerR, which enable the detection of specific ROS and orchestrate the staphylococcal defensive gene expression programs (59). In *S. aureus*, PerR functions as a principal peroxide sensor and belongs to the ferric uptake regulator (Fur) family of metalloregulatory proteins, which includes Fur and Zur (60). These regulators require metal cofactors for activity, underscoring the metal-dependent nature of oxidative stress sensing. In our analysis, PerR emerged as a key molecular determinant of oxidative stress resistance in *S. aureus*, a finding confirmed by experimental testing. The *per* mutant exhibited a significant fitness defect in serum supplemented with 0.25- and 0.5 mM hydrogen peroxide ($H_2O_2$), with the most pronounced impairment observed after 20 h of exposure to 0.5 mM $H_2O_2$ (Fig. 3F; Fig. S5D). Exposure to serum also elicited significant production of staphyloxanthin, encoded by the *crt* operon, and whose production is under the control of PerR (Fig. 3G; Fig. S5E). These results underscore the critical role of PerR in facilitating *S. aureus* survival under oxidative stress conditions, as encountered by *S. aureus* in serum, highlighting its importance in maintaining bacterial fitness in ROS-rich host environments.

## Integrated metabolic responses of *S. aureus* to human serum

To conceptualize the dynamic remodeling of *S. aureus* metabolism in response to serum, we constructed a scaffolded metabolic network based on curated literature (61–74). This network was overlaid with the significant changes in key enzymes identified through single-omics analysis (genes upregulated in orange, downregulated in light blue) and by DIABLO integration (genes and metabolites in red), in combination with network analyses (Fig. 4; Tables S2 to S5). This integrative approach revealed that *S. aureus* mounts a broad and adaptable metabolic response to the challenges posed by the serum environment, particularly in the utilization of diverse carbon sources including

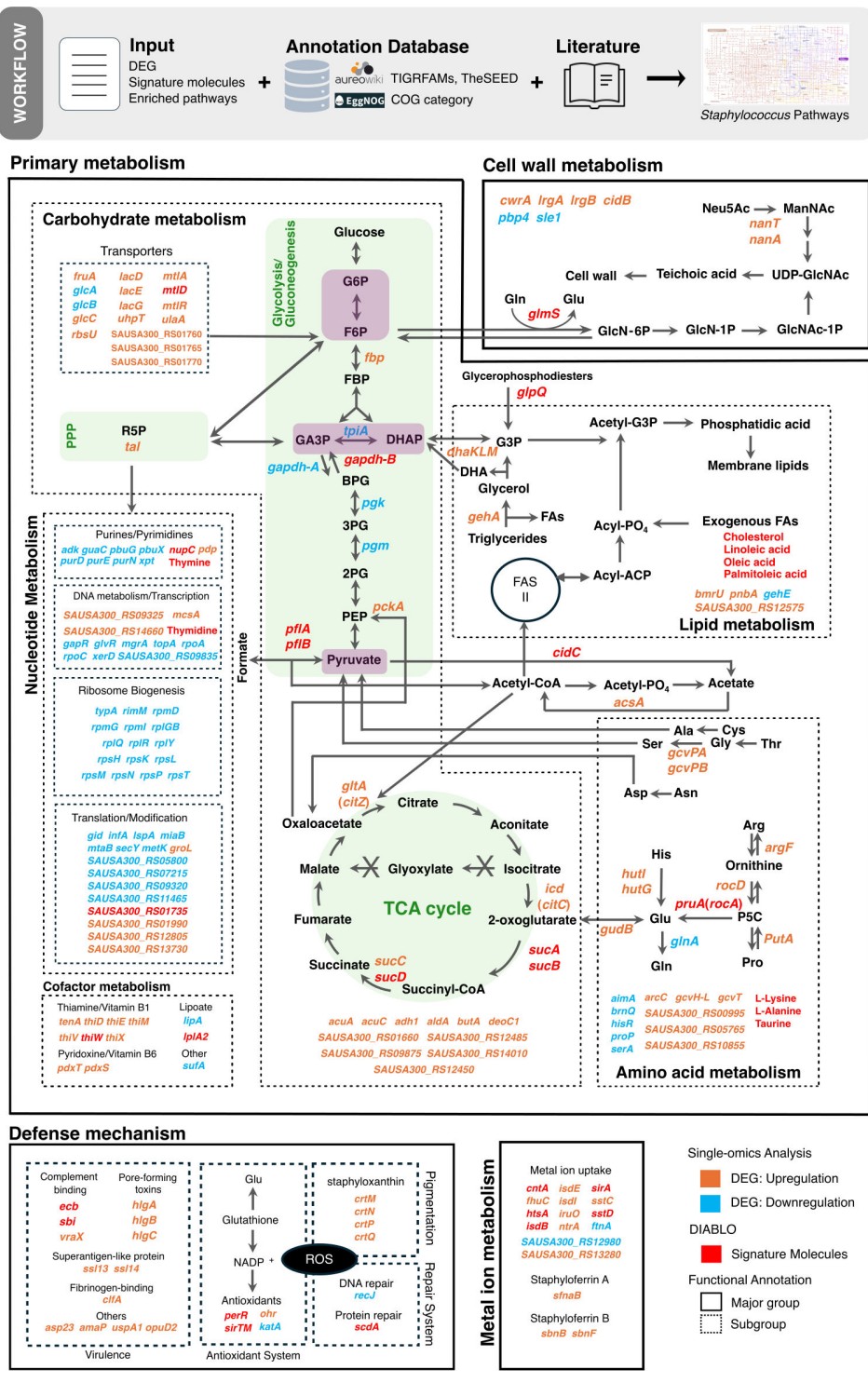

**FIG 4** Integrated metabolic and virulence adaptation of *S. aureus* in response to serum. Systems-level metabolic reprogramming of *S. aureus* upon exposure to serum, integrating single- and multi-omics data. The depicted key metabolic pathways include glycolysis/gluconeogenesis (upright rectangle shaded in light green), the pentose phosphate pathway (PPP), the tricarboxylic acid (TCA) cycle (circle shaded in light green), nucleotide biosynthesis, ribosome biogenesis, thiamine (vitamin B1) biosynthesis and transport, cell wall biosynthesis, amino acid metabolism and lipid metabolism. Host-responsive processes, such as immune evasion, metal ion acquisition, and oxidative stress responses, are also represented. The name of the enzyme is colored to indicate changes in gene expression following serum exposure identified by single omics (compared to RPMI): orange indicates increased expression, light blue indicates decreased expression. Variables predicted by DIABLO (multi-omics analysis) are in red, indicating their status as signature molecules of *S. aureus* serum response.

carbohydrates, amino acids, and lipids. Multi-omics data (Fig. S7A through D) highlighted consistent upregulation of gluconeogenic and TCA cycle enzymes at both transcript and protein levels, indicating a metabolic shift toward energy-efficient carbon utilization upon exposure to serum.

Carbohydrate metabolism was organized around three key metabolic nodes—glucose-6-phosphate/fructose-6-phosphate (G6P/F6P), glyceraldehyde-3-phosphate/dihydroxyacetone phosphate (GA3P/DHAP), and pyruvate (Fig. 4, purple nodes embedded in glycolysis/gluconeogenesis)—which serve as convergence points for nutrient assimilation. The upregulation of carbohydrate transporter genes (*uhpT, mtlA, fruA, glcC, lacE, SAUSA300_RS01760, rbsU*) in serum aligns with our findings that saccharides, including the hexose monosaccharides α-D-glucose, D-fructose, D-mannose, and disaccharides D-maltose and sucrose, significantly enhanced the metabolic activity of *S. aureus* (Fig. 3C; Fig. S6A). Notably, saccharides, such as G6P, fructose, and mannose, but not glucose, have been shown to influence virulence gene expression (75). Our transcriptomic data support this, revealing increased expression of γ-hemolysins genes (*hlgA, hlgB, hlgC*) and immune evasion genes (*ecb, sbi, vraX*) across five *S. aureus* strains in serum. In particular, strain BPH2986 (ST8) exhibited strong upregulation of *lukF-PV* and *lukS-PV* genes, which encode Panton-Valentine leucocidins (Fig. S8A). These findings suggest that carbohydrates, such as fructose and mannose, present in serum support the metabolic activity of *S. aureus* while increasing virulence gene expression, in contrast to the glucose-rich RPMI.

Amino acids and peptides from serum further contribute to the metabolic adaptability of *S. aureus* by entering carbohydrate metabolism and the TCA cycle through key intermediates, including pyruvate, 2-oxoglutarate, and oxaloacetate. These entry points define three functional groups of amino acids: (i) pyruvate-yielding, (ii) 2-oxoglutarate-yielding, and (iii) oxaloacetate-yielding (Fig. 4). Our transcriptomic data revealed significant upregulation of enzymes involved in amino acid catabolism, including *pckA* (encoding pyruvate carboxykinase) and *gudB* (encoding glutamate dehydrogenase), consistent with elevated intracellular levels of phosphoenolpyruvate (PEP) and 2-oxoglutarate (Fig. S7C through E). Genes encoding enzymes for the degradation of glutamate (*gudB*), proline (*pruA/rocA, putA*), arginine (*rocD/argD*), and histidine (*hutI, hutG*) were also upregulated, indicating preferential catabolism of these amino acids during serum exposure. These findings are consistent with previous reports identifying glutamate and its precursors as major carbon sources supporting *S. aureus* proliferation (65), positioning glutamate as a central amino acid in the adaptive response to serum.

Exposure to human serum triggered a marked upregulation of *S. aureus* genes involved in lipid and phospholipid degradation, including *glpQ* and *gehA*. This suggests an enhanced capacity in using host-derived lipids—such as cholesterol, linoleic acid (C18:2), and oleic acid (C18:1)—as nutrient sources. These lipids are metabolized into glycerol-3-phosphate (G3P), which can then be funneled into the carbohydrate metabolism via the GA3P/DHAP node. Interestingly, our transcriptomic data revealed downregulation of *tpiA*, which encodes triosephosphate isomerase, an enzyme responsible for the reversible interconversion of GA3P and DHAP. This suggests a potential bottleneck at this metabolic junction, which may constrain glycolytic flux and influence the efficiency of carbon utilization from lipids present in serum.

Counteracting innate immune defenses is essential for *S. aureus* survival during bloodstream infection. Upon exposure to human serum, *S. aureus* exhibited robust transcriptional upregulation of a broad range of virulence-associated genes involved in immune evasion, host colonization, and pathogenesis. These include genes encoding complement-evasion proteins (*ecb, sbi, vraX*), pore-forming cytotoxins (*hlgA, hlgB, hlgC*), superantigen-like proteins (*ssl13, ssl14*), and the fibrinogen-binding adhesin *clfA*. This coordinated immune evasion strategy underscores the pathogen's capacity to rapidly adapt to immune pressures and enhance its invasive potential in the bloodstream.

Serum exposure also elicited a pronounced oxidative stress response in *S. aureus*. DIABLO analysis identified *perR*, which encodes the peroxide-sensing transcriptional

regulator PerR, as a key molecular signature of staphylococcal adaptation to serum. Consistent with this, genes involved in carotenoid biosynthesis—*crtM, crtN, crtQ*—were upregulated, suggesting enhanced anti-oxidative defense through increased production of staphyloxanthin, a carotenoid pigment known to protect against ROS (76). In parallel, elevated expression of *scdA*, a gene involved in iron-sulfur cluster repair, indicates activation of protein maintenance pathways essential for preserving enzymatic and cellular functions under oxidative stress (37, 77).

Our integrative approach has highlighted the high adaptive capacity of *S. aureus* to withstand human serum exposure. Through the coordinated activation of nutrient acquisition systems, oxidative stress defenses, and virulence-associated mechanisms, *S. aureus* enhances its ability to survive and persist during systemic infection.

## DISCUSSION

While a leading cause of lethal bacteremia, *S. aureus* must contend with the nutrient-limited and immunologically hostile conditions of the bloodstream to establish infection and cause disease. In the bloodstream, the bacterium uses a large panoply of virulence factors to evade clearance by cellular and humoral immunity (1, 78, 79). Notwithstanding bactericidal immune cells patrolling the bloodstream, serum remains a challenging environment for *S. aureus* due to the paucity of free forms of essential nutrients, which are often complexed to large proteins (e.g., iron ions complexed to transferrin) (80). Several studies have shown that *S. aureus* has evolved multiple mechanisms, unrelated to its cytotoxicity, to overcome the nutritional constraints and acellular insults of the serum (16). While informing on discrete *S. aureus* nutritional stress responses, these mechanisms were studied in isolation and did not consider the concomitant adaptive responses to the serum environment. Building on our previous work showing that four diverse blood-invasive pathogens, including *S. aureus*, remodeled their cell wall to adapt to serum (18), we used a multi-omics approach to define the shared serum-adaptive response of *S. aureus* across five clinical isolates. Such an approach offers a systems-level view of microbial adaptation in response to environmental stimuli, by linking molecular changes across genomic, transcriptomic, proteomic, and metabolomic layers (81–83). Using integrative machine learning-based multivariate analyses, differential expression profiling, and network-based approaches, our study identified staphylococcal reprogramming in cell wall biosynthesis, carbon metabolism, iron transport, and defense mechanisms in response to serum.

The experimental validation of the dominant outputs identified by our integrative framework confirmed the importance of *gapdhB*, *sucA*, *sirA*, *sstD*, and *perR* genes for staphylococcal fitness in serum (Fig. 3). These results align with the positive co-variation patterns observed in the DIABLO model, suggesting a functional interdependence among the pathways involving these genes (Fig. 1E). While critical for optimal fitness in serum, *gapdhB* and *sucA* have distinct functions in carbon metabolism. Metabolic profiling revealed that hexose saccharides and derivatives (e.g., α-D-glucose, D-glucosamine) enhanced the metabolic activity in WT and *gapdhB* and *sucA* mutants, through glycolysis, gluconeogenesis, PPP, and TCA cycle. While methyl pyruvate partially restored bacterial respiration in the *sucA* mutant, likely via conversion to acetyl-CoA and oxaloacetate (84), the *gapdhB* mutant remained metabolically impaired, reflecting incomplete gluconeogenesis and insufficient PPP intermediate generation (Fig. 3B; Fig. S6). While highlighting the importance of gluconeogenesis for *S. aureus* during serum exposure, these results further underscore the non-redundant roles of GAPDH homologs (*gapdhA*, *gapdhB*) in maintaining metabolic flux (29). Interestingly, pyruvate was shown to activate *S. aureus* regulatory networks controlling the production of leucocidins, thereby potentiating the cytotoxicity of *S. aureus* (85). Conversely, exposure to glyoxylate and itaconate inhibited *S. aureus* metabolism (Fig. 3C; Fig. S6). The bacterium does not use the glyoxylate shunt (86), rendering glyoxylate toxic due to metabolic imbalance. Moreover, glyoxylate is limited in serum, as humans lack an active glyoxylate cycle (65). Itaconate, an immunometabolite produced by macrophages, further exacerbates stress

by directly inhibiting glycolysis (86, 87). These findings highlight the potential of *S. aureus* metabolic vulnerabilities that could be exploited therapeutically, such as inhibitors of the bacterial glycolysis and the TCA cycle (88). However, itaconate was shown to increase tolerance to aminoglycosides, limiting its potential as adjunct therapeutic (89). We constructed a metabolic map of *S. aureus* that integrates the predicted signature molecules elicited/repressed in serum (Fig. 4; Table S4). This illustrates the flexibility of *S. aureus* in using diverse carbon sources in serum, such as hexose saccharides, amino acids yielding 2-oxoglutarate (e.g., Glu, Pro, Arg, His), and host lipids and fatty acids (e.g., cholesterol, linoleic acid [C18:2], and oleic acid [C18:1]) to support its survival. Hexose saccharides have also been shown to regulate virulence gene expression (75). Peptides and amino acids present in serum contribute to *S. aureus* metabolism by entering glycolysis via pyruvate or intermediates of the TCA cycle (e.g., 2-oxoglutarate, oxaloacetate). Transcriptomic analysis revealed significant upregulation of key enzymes involved in amino acid catabolism, including pyruvate carboxykinase (*pckA*) and glutamate dehydrogenase (*gudB*), as well as increased levels of phosphoenolpyruvate (PEP) and 2-oxoglutarate (Fig. S7C through E). Notably, we found a significant increase in gene expression of enzymes (e.g. *gudB*, *pruA* [*rocA*], *putA*, *rocD* [*argD*], *hutl*, and *hutG*) that convert amino acids (e.g., Glu, Pro, Arg, His) to 2-oxoglutarate in amino acid catabolism, indicating the usage of these amino acids by the bacteria in serum. This is consistent with the observation that glutamate, along with the amino acids that can be converted into glutamate (Pro, Arg, and His), provides most of the carbon required for bacterial growth (65). Several upregulated metabolic pathways facilitating glutamate production, driven by *pruA* (converting proline to glutamate) and *putA* (proline dehydrogenase), underscore glutamate as a key intermediate metabolite in the adaptive response of *S. aureus* to serum. Concurrently, the downregulation of glutamine synthesis, through decreased expression of *glnA*, suggests a reduction in the utilization of glutamate. Excess levels of glutamate did not significantly influence the respiration or growth of JE2 WT (Table S4), although several studies have highlighted the important role of glutamate in protecting bacteria from cell death under stress (90–93).

Iron is an important determinant for *S. aureus* fitness (80). Our analyses confirmed the role of multiple iron acquisition mechanisms in iron-limited host environments such as serum. Interestingly, mutants lacking *htsA* and *isdB*, both involved in iron acquisition, displayed growth rates comparable to that of JE2 WT, suggesting some iron uptake system redundancy (Fig. 3E). This observation is consistent with prior research reporting the upregulation of iron acquisition systems, such as Isd and SirABC, in *S. aureus* cultured in serum or blood (94). Further studies are necessary to understand the specificity of iron acquisition modules in iron-limited environments in the context of bacterial metabolic reprogramming during bloodstream infection.

Resistance to oxidative stress in serum was also highlighted by our analyses and confirmed experimentally. Staphyloxanthin is a carotenoid pigment produced by *S. aureus* that enhances resistance to oxidative stress (76). The biosynthesis of staphyloxanthin is encoded by the *crtOPQMN* operon, primarily activated by σB, and its expression is influenced by redox-responsive regulators. PerR, a peroxide-sensing transcriptional repressor identified by DIABLO as a response signature to serum, plays a pivotal role by controlling genes that detoxify ROS, such as *katA*, which encodes catalase. Loss of PerR function disrupts oxidative stress homeostasis, which can alter σB activity, leading to reduced production of staphyloxanthin (Fig. 3G; Fig. S5E). Thus, our data confirmed that PerR is a key player linking peroxide sensing to virulence factor regulation upon serum exposure (43, 60).

Serum exposure also engaged *S. aureus* defense mechanisms against acellular immunity. We observed upregulation of virulence-related genes, including complement-binding proteins (e.g., *ecb*, *sbi*, *vraX*) (28, 95, 96), pore-forming toxins (e.g., the γ-hemolysins *hlgA*, *hlgB*, *hlgC*) (97), superantigen-like proteins (*ssl13*, *ssl14*) (98), and the fibrinogen-binding protein *clfA* (99) (Fig. 4; Fig. S8). Such responses would prime *S. aureus* against clearance by immune cells. Mechanisms for evasion from host antibodies

and the complement system were also engaged, including the immunoglobulin-binding protein (Sbi) and complement convertase inhibitor (Ecb) (28, 100). *S. aureus* sensing and responses to host oxidative stress were also identified as important for serum exposure survival (40, 43, 47, 60). Future studies examining how serum promotes *S. aureus* biofilm formation on host-like surfaces could provide deeper insight into the roles of ClfA, Ecb, and other adherence factors during bloodstream infections.

## Conclusion and perspectives

This study exploited multi-omics analyses, including machine learning-based approaches, DE analysis, and network analysis, to uncover the major *S. aureus* features engaged in response to human serum. Our findings highlight the parallel activation of nutrient acquisition systems, oxidative stress defenses, and virulence-associated mechanisms, underlining the strong adaptive capacity of *S. aureus* to withstand the multifactorial challenges imposed by human serum. While serum is acellular and does not reflect the breadth of host-pathogen interactions during bloodstream infections, serum can be an informative proxy for understanding how invading bacteria survive in blood. The key staphylococcal pathways identified here could be further dissected to explore staphylococcal adaptive responses and identify prospective bacterial (and host) targets for new therapeutic approaches.

## Limitations of the study

The integrative analysis performed here was highly sensitive and detected subtle patterns linking different omic data sets. However, the metabolic flexibility and functional redundancy within key bacterial pathways—particularly central carbon metabolism and iron acquisition—are likely not limited to the individual contributions of specific genes, thus making genes contributing to these pathways difficult to confirm experimentally using single isogenic mutants. Furthermore, the individual contributions of metabolites, such as citrate, an intermediate of the TCA cycle, can repress the activity of the transcriptional activity of genes involved in iron homeostasis (101), underscoring the cross-talks between pathways that could alter single-omic readouts and, consequently, multi-omics analyses. This study used commercially sourced serum for phenotypic confirmation of the staphylococcal signatures. While such a reagent offers more control, such as batch control, the concentrations of carbohydrates, metal ions, and ROS are unknown, which may influence the significance of phenotypic outputs. Moreover, the data sets analyzed here were generated using pooled human sera (Lifeblood, Melbourne, Australia), while our phenotypic confirmation experiments were conducted using commercially sourced, batch-controlled human sera (Sigma Aldrich). The difference in serum processing may explain the discrepant phenotypes obtained for the siderophore-independent *isd* genes involved in iron scavenging from heme between our current study and the study that generated the data sets (18). While RPMI, closely mimicking the nutrient-limited milieu encountered by *S. aureus* during host infection (102), was used as the comparator to serum in our studies (18; present study), bacterial omics analyses generated from exposure to alternative media, such as the human plasma-like media (HPLM) (103), enriched in host-like metabolites, could refine the staphylococcal signatures in future analyses.

## MATERIALS AND METHODS

### *S. aureus* genome and pangenome analyses

The complete genomes of five clinical *S. aureus* strains, BPH2760, BPH2819, BPH2900, BPH2947, BPH2986 (18), and of the strain USA300 JE2 were downloaded from the NCBI database (https://www.ncbi.nlm.nih.gov/bioproject/PRJEB29881/, https://www.ncbi.nlm.nih.gov/bioproject/PRJNA381486/).

Genomic analysis was conducted using Bohra v2.3.6 (https://github.com/MDU-PHL/bohra). Bohra provided information, such as SNPs, phylogeny, multi-locus sequence typing (MLST), resistome, virulome, and pan-genome. We used Phandango to visualize the pangenome profiles (https://jameshadfield.github.io/phandango/#/).

The AureoWiki database (24) was used to annotate *S. aureus* genes/proteins and to define orthologous relationships through a reciprocal best hit (RBH) search using MMseqs2 with default parameters (https://github.com/soedinglab/MMseqs2) between the five studied genomes (FAA format) and the *S. aureus* reference genome and proteome (USA300_FPR3757_NCBI_2017 and USA300_FPR3757_NCBI_UniProt_2013).

## Confirmation of *S. aureus* transposon mutants' genome sequence

We used *S. aureus* transposon mutants of signature genes identified by our analytical pipeline to verify their fitness in serum. These mutants were selected from the Nebraska transposon library (NTL), including NE275, NE400, NE547, NE665, NE1102, NE1343, and NE1767 (104). The mutants are expected to contain a precise and non-polar transposon insertions, without additional mutations. All mutants were genome sequenced on the Oxford Nanopore platform using Min114 flow cells. Base calling, demultiplexing, and adapter removal were performed in real time using MinKNOW. Mutant NE1102 was sequenced on a GridION and processed using Guppy. Remaining mutants were sequenced on a MinION and processed using Dorado. FASTQ reads were assembled using Trycycler version 0.5.3 (105). The assembled genomes were compared to the cognate wild-type reference genome using the Mauve whole-genome aligner plugin tool in the Geneious Prime (version 2022.2.2) (106).

## Serum treatment

We used batch-controlled human serum (Male AB plasma, Sigma Aldrich) to reduce variations arising from different donors. Human serum was clarified from lipids by centrifugation at $3,000 \times g$ for 5 min at 4°C. We further analyzed the protein content of clarified serum (Fig. S9). Human serum samples (pre- [untreated {U}] and post- [treated {T}] centrifugation 5 min at $3,000 \times g$) were diluted 1:50 in sterile PBS, then boiled 5 min with 5 mM DTT in 1× protein loading buffer. Equivalent amounts of proteins were loaded on a Bolt 4%–12% Bis-Tris SDS PAGE gel and separated using MOPS-SDS running buffer (ThermoFisher, USA) at 100 volts for 1 h. The gel was stained with Coomassie blue overnight, destained, and imaged using a GE Amersham Imager 800 (GE Life Sciences, UK).

## *S. aureus* fitness in serum

Overnight cultures of *S. aureus* JE2 (WT) and NTL mutants of *gapdhB*, *sucA*, *htsA*, *sirA*, *sstD*, *isdB*, and *perR* were grown from single colonies in 10 mL brain heart infusion media (BHI; BD BACTO) at 37°C with orbital shaking. Overnight cultures were centrifuged for 5 min at $5,000 \times g$, and pellets were washed twice with sterile PBS. The cells were then standardized to an OD at 600 nm = 0.5 (0.5 UOD) in RPMI (Gibco). Bacteria equivalent to 0.05 UOD were inoculated in 180 µL of BHI, RPMI, or 50% serum (diluted in RPMI) in a clear 96-well plate (Costar) in triplicate to a final volume of 200 µL. Plates were incubated at 37°C in ClariostarPLUS microplate reader (BMG Labtech), and bacterial growth was measured every 10 min for 20 h at 600 nm.

## Phenotypic analysis of *S. aureus* carbohydrate metabolism

Phenotypic profiling was performed using the Biolog Phenotype MicroArray (PM) system with PM1 and PM2 plates (Biolog Inc., Hayward, CA, USA) to determine the *S. aureus* fitness in the presence of 190 distinct carbon sources. Overnight *S. aureus* cultures were grown in BHI at 37°C with shaking. Bacterial suspensions were prepared according to the manufacturer's protocol for Gram-positive organisms. The inocula of *gapdhB*, *sucA*, and parental WT strain were loaded into PM1 and PM2 plates and incubated at 37°C for

24 h in the Biolog Odin system, which enables the simultaneous monitoring of growth and respiration (by redox dye reduction). Metabolic activity was assessed by measuring the reduction of tetrazolium-based redox dyes, which serve as indicators of electron transport chain activity in response to specific carbon sources. Dye reduction results in the formation of purple formazan, with absorbance recorded every 20 min at 590 nm. The area under the curve (AUC) for each well was calculated to quantify metabolic activity, enabling comparative analysis of carbon source utilization across conditions or strains.

## Quantification of staphyloxanthin

Overnight cultures of *S. aureus* JE2 (WT) and NTL mutants of *perR* and *crtM* were grown from single colonies in 25 mL BHI at 37°C with orbital shaking. The cultures were standardized to an OD at 600 nm = 25. These were then centrifuged for 15 min at 5,000 × *g*, and pellets were washed twice with sterile PBS. Bacteria were incubated in 700 µL of RPMI or in 50% serum (diluted in RPMI) at 37°C with orbital shaking for 5 h. The bacteria were then centrifuged for 5 min at 5,000 × *g*, and the supernatant was discarded. The pellets were resuspended in 450 µL of methanol, transferred in 1.5 mL tubes, and incubated at 55°C for 15 min with shaking. Tubes were centrifuged for 2 min at 13,000 RPM, at room temperature. One hundred microliters of supernatant from each tube was added to a clear 96-well plate (Costar) in triplicate, and the extracted staphyloxanthin was quantified at 450 nm using a ClariostarPLUS microplate reader (BMG Labtech). Staphyloxanthin production was calculated by subtracting each data point from the mean absorbance of *crtM* to normalize different condition backgrounds (RPMI and serum). Replicates were generated from four different cultures for each strain and assessed in technical triplicate across two distinct commercial serum batches.

## Principal component analysis

PCA was independently performed on each omics data set—transcriptomics, proteomics, GC-MS metabolomics, and LC-MS metabolomics—to explore global data structure and identify the primary sources of variation between serum- and RPMI-exposed samples. PCA was conducted using the pca() function from the mixOmics package (v6.24.0) in R (v4.2.3), with data scaled to unit variance. The optimal number of components was determined using the tune.pca() function, which indicated that the first three components captured approximately 50% of the total variance across data sets.

## Multi-omics data integration

We implemented the MixOmics data analytic pipeline known as multiblock (s)PLS-DA or DIABLO, a multivariate statistical method for classifying and separating different categorical classes (19, 20). This supervised multi-omics integrative method adds sparsity to the model, improving result interpretability by selecting the most relevant variables and setting the coefficients of others to zero. DIABLO treats each omics data set as a separate "block" and identifies co-variation patterns across these blocks to determine interactions between, and changes by, biological molecules under various conditions. The DIABLO model was designed with a selected design matrix weight of 0.5 to balance discrimination between conditions and correlation across omics layers, ensuring that feature selection prioritized both inter-omics agreement and RPMI-serum separation. An initial model trained with five components revealed that the first component alone was sufficient to clearly discriminate between serum and RPMI conditions. To assess model robustness and guide component selection, we performed leave-one-out (LOO) cross-validation, which confirmed that a single component provided perfect classification performance. To optimize feature selection for each omics layer, we performed parameter tuning using the tune.block.splsda function. A grid of candidate values for the number of features to retain per block was defined: 20–100 (step size 20) for transcriptomics and proteomics and 10–100 (step size 10) for both metabolomics data

sets (GC-MS and LC-MS). The model was tuned using 5-fold cross-validation repeated 10 times and centroid distance as the classification metric. The number of features retained per block was 20 for transcriptomics, 20 for proteomics, 10 for GC-MS, and 10 for LC-MS, totaling 60 features. Using these parameters, the final DIABLO model was built using the block.splsda function with one component (as determined during component tuning) (19, 20).

## Enrichment analysis

To identify staphylococcal pathways prominently represented following exposure to serum, feature lists derived from all omics data underwent PEA using ORA and GSEA. PEA was performed with the package clusterProfiler v4.6.2. The obtained feature lists were mapped to the associated biological annotation terms (e.g., GO terms or KEGG pathways), and statistical tests for enrichment probability (enrichment $P$-value) were calculated to determine the enrichment of the feature sets. In ORA, the preselected features were filtered by a criterion of log2FC > 1 and $P$-value < 0.05. For GSEA, all features were ranked based on their log2FC values, so that features with the largest positive and negative expression changes are positioned at opposite ends of the ranked list, while those with smaller or no changes are located near the middle.

## Network analysis

The biological relationships among the list of enriched pathways were identified by selecting significant pathways with $P$.adjust < 0.05 (except for metabolomic data sets; $P$-value < 0.05). Further network analysis, using the pairwise_termsim function in the enrichplot package in R, calculated the pairwise similarity of the enriched terms using Jaccard's similarity coefficient or the similarity of gene subsets sharing between pathways. Conversion to enrichment map to dataframes provided similarity scores and number of strains sharing individual enriched pathways, represented as edge and node , respectively, for networks' visualization.

## ACKNOWLEDGMENTS

This research was supported by the Development and Promotion of Science and Technology Talents Project (DPST) from the Thai Government (W.M.), the National Health and Medical Research Council of Australia to B.P.H. (GNT1196103), T.P.S. (GNT1194325), and A.H., S.G., and R.G. (GNT2018880).

## AUTHOR AFFILIATIONS

[1]Department of Microbiology and Immunology, The University of Melbourne at the Peter Doherty Institute for Infection and Immunity, Melbourne, Victoria, Australia

[2]Centre for Pathogen Genomics, The University of Melbourne, Melbourne, Victoria, Australia

[3]Victorian Infectious Diseases Service, Royal Melbourne Hospital, The Peter Doherty Institute for Infection and Immunity, Melbourne, Victoria, Australia

[4]Melbourne Integrative Genomics, School of Mathematics and Statistics, University of Melbourne, Melbourne, Victoria, Australia

[5]Microbiological Diagnostic Unit Public Health Laboratory, Department of Microbiology and Immunology, University of Melbourne, Doherty Institute, Melbourne, Victoria, Australia

## AUTHOR ORCIDs

Warasinee Mujchariyakul  http://orcid.org/0000-0001-7994-5979
Calum J. Walsh  http://orcid.org/0000-0003-3113-9534
Stefano Giulieri  http://orcid.org/0000-0001-5366-1943
Kim-Anh LêCao  http://orcid.org/0000-0003-3923-1116

Timothy P. Stinear  http://orcid.org/0000-0003-0150-123X
Benjamin P. Howden  http://orcid.org/0000-0003-0237-1473
Romain Guérillot  http://orcid.org/0000-0001-9915-1420
Abderrahman Hachani  http://orcid.org/0000-0001-8032-2154

## FUNDING

| Funder | Grant(s) | Author(s) |
|---|---|---|
| Thai Development and Promotion of Science and Technology Talents Project | | Warasinee Mujchariyakul |
| National Health and Medical Research Council | GNT2018880 | Abderrahman Hachani |
| | | Romain Guérillot |
| | | Stefano Giulieri |
| National Health and Medical Research Council | GNT1196103 | Benjamin P. Howden |
| National Health and Medical Research Council | GNT1194325 | Timothy P. Stinear |

## AUTHOR CONTRIBUTIONS

Warasinee Mujchariyakul, Conceptualization, Data curation, Formal analysis, Funding acquisition, Investigation, Methodology, Project administration, Software, Supervision, Validation, Visualization, Writing – original draft, Writing – review and editing | Calum J. Walsh, Conceptualization, Data curation, Formal analysis, Investigation, Methodology, Resources, Software, Supervision, Writing – review and editing | Stefano Giulieri, Conceptualization, Formal analysis, Funding acquisition, Investigation, Project administration, Supervision, Writing – review and editing | Cameron Cramond, Investigation, Writing – review and editing | Kim-Anh LêCao, Conceptualization, Formal analysis, Funding acquisition, Investigation, Resources, Software, Supervision, Visualization, Writing – review and editing | Timothy P. Stinear, Conceptualization, Formal analysis, Funding acquisition, Investigation, Resources, Software, Supervision, Visualization, Writing – review and editing | Benjamin P. Howden, Conceptualization, Formal analysis, Funding acquisition, Investigation, Project administration, Supervision, Writing – review and editing | Romain Guérillot, Conceptualization, Data curation, Formal analysis, Investigation, Methodology, Resources, Software, Supervision, Writing – review and editing | Abderrahman Hachani, Conceptualization, Data curation, Formal analysis, Funding acquisition, Investigation, Methodology, Project administration, Software, Supervision, Validation, Visualization, Writing – original draft, Writing – review and editing

## DATA AVAILABILITY

All codes are available on GitHub (https://github.com/warasinee/Multiomics_Analyses_2024). Oxford Nanopore reads of all strains and mutants generated are deposited in NCBI SRA (BioProject ID PRJNA1284587). Transcriptomics data have been deposited in NCBI GEO under accession numbers GSE152833, GSE152834, GSE152835, GSE152837, and GSE152838. Proteomics data have been deposited in PRIDE under accession numbers PXD016504 and PXD020791. Metabolomics data have been deposited in Metabolights under accession number MTBLS1898.

## ADDITIONAL FILES

The following material is available online.

### Supplemental Material

**Supplemental Figures (mSystems01183-25-s0001.pdf).** Figures S1-S9.
**Table S1 (mSystems01183-25-s0002.xlsx).** DIABLO first component variables.
**Table S2 (mSystems01183-25-s0003.xlsx).** Common DE lists.

**Table S3 (mSystems01183-25-s0004.xlsx).** Detailed information of network analysis, related to Fig. 2 and S3.
**Table S4 (mSystems01183-25-s0005.xlsx).** Biolog data.
**Table S5 (mSystems01183-25-s0006.xlsx).** Composition of RPMI and serum.

## Open Peer Review

**PEER REVIEW HISTORY (review-history.pdf).** An accounting of the reviewer comments and feedback.

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
