## [Reviewer comments · mSystems]

Integrated multi-omics reveals coordinated *Staphylococcus aureus* metabolic, iron transport and stress responses to human serum

Warasinee Mujchariyakul, Calum Walsh, Stefano Giulieri, Cameron Cramond, Kim-Anh Lê Cao, Timothy Stinear, Benjamin Howden, Romain Guérillot, and Abderrahman Hachani

Corresponding Author(s): Abderrahman Hachani, The University of Melbourne

Review Timeline:

Submission Date:	August 9, 2025
Editorial Decision:	September 18, 2025
Revision Received:	November 11, 2025
Accepted:	December 17, 2025

Editor: Marnix Medema

Reviewer(s): Disclosure of reviewer identity is with reference to reviewer comments included in decision letter(s). The following individuals involved in review of your submission have agreed to reveal their identity: Oscar Quijada Pich (Reviewer #1)

Transaction Report:

DOI: <https://doi.org/10.1128/msystems.01183-25>

Re: mSystems01183-25 (**Integrated multi-omics reveals coordinated *Staphylococcus aureus* metabolic, iron transport and stress responses to human serum**)

Dear Dr. Abderrahman Hachani:

In general, both reviewers are positive, but especially reviewer #1 has several recommendations that could further strengthen the paper. It would be great if you could consider carefully also the suggestions for further experiments, and carry them out if feasible or clearly argue to me and the reviewers if/why they are not needed.

Revision Guidelines

Sincerely,
Marnix Medema
Editor
mSystems

Reviewer #1 (Comments for the Author):

This is a thorough manuscript that makes an important contribution to understanding how *Staphylococcus aureus* adapts to the hostile environment of human serum. The integration of transcriptomic, proteomic, and metabolomic data using the DIABLO

framework is innovative and provides a systems-level view of bacterial adaptation. The authors strengthen their findings with experimental validation of key genes, demonstrating the biological relevance of their multi-omics signatures.

That said, there are several areas where the manuscript could be improved. Some sections of the results and discussion are dense, with long sentences that may be difficult for readers less familiar with multi-omics. Additionally, the explanation of the DIABLO framework could be simplified or accompanied by an intuitive description to improve accessibility.

Methodological details are generally clear, but some aspects could be expanded to ensure reproducibility—for example, serum preparation and fitness assays. It would also help to consistently indicate the number of biological replicates in the figure legends.

Major Points

1. Simplify DIABLO explanation

The DIABLO framework is complex; a more intuitive description or schematic would make the analysis more accessible to readers unfamiliar with multivariate approaches.

2. Use of RPMI as the only comparator

RPMI is used as the sole baseline for serum exposure. While defined and reproducible, it differs substantially from serum in protein content, nutrients and host factors such as heme, albumin and complement. Some observed bacterial responses may therefore reflect media differences rather than true host-specific adaptation. The authors should justify why RPMI alone is appropriate or clarify the limitations of this approach. Consider whether heat-inactivated serum or additional controls could strengthen the physiological relevance.

3. Use of commercial serum

As noted by the authors, commercial serum may not fully recapitulate physiological serum composition, which could influence pathway activation and mutant phenotypes (e.g., *isdB*). This limitation should be emphasized in the discussion.

4. Study *isdB* role under heme-containing conditions

Reassessing *isdB* mutant fitness in serum supplemented with hemoglobin or heme (physiological range), or in lysed RBC-containing serum, would help explain the lack of phenotype in commercial serum and clarify the context-dependent utility of the *Lsd* system.

5. Quantify oxidative stress and antioxidant responses directly

Measure intracellular ROS (e.g., H₂DCFDA), catalase/peroxidase activities and staphyloxanthin levels (HPLC or spectrophotometry) in WT and *perR* mutants with and without serum or H₂O₂. This would provide functional support for the role of *PerR* and carotenoid pathways in oxidative protection beyond growth curves.

6. Biofilm and adhesion assays on serum-coated surfaces

Evaluate whether serum exposure alters adherence or biofilm formation on host-like surfaces and test relevant mutants (e.g., *ecb*, *sbi*, *clfA*). This would link metabolic or vulnerability changes to colonization-relevant phenotypes.

Minor Points

Line 121: "approximately": please indicate variance explained by PC1 for each dataset

Line 135: Pearson correlations >0.95 with only 5 isolates are surprisingly high. Consider adding p-values. Clarify whether replicates are treated as independent samples.

Figure 1E: "Proteome" should not be italicized; capitalize the first letter.

Serum composition: consider attempting to determine this more precisely.

Supplementary material: some plots are difficult to read (likely a PDF rendering issue).

Line 216: missing period after "(Table S2 and S3)".

Line 599: link contains a space and does not work.

Reviewer #2 (Comments for the Author):

This paper by Mujcharyakul and colleagues builds on the field of human serum effects on bacteria. Their prior work with high priority pathogens provide the groundwork to focus on *S. aureus* across different sequence types. This works studies 5 highly relevant sequence types from clinical *S. aureus*, using the NTML for target validation. This is one of the most comprehensive studies to date, encompassing multi-omic data integration to identify adaptive responses and pathways in serum. The results are largely expected given the nutrient limitation in this environment, but it does provide specific connected pathways and targets. The authors mention these could be targets for future therapeutic development, but the manuscript does a poor job of providing details or examples of how this could be employed. Metabolic interventions/inhibitors have not been highly successful in development (at least when used alone), so the authors could also state that in the limitation and/or discussion. Finally, given the importance of the findings, it may be worthwhile to reflect their findings on the burgeoning field of using alternative media (e.g. mammalian cell culture growth) for susceptibility testing of antibiotics, which may be more akin to the serum environment with nutrient availability.

November 5th, 2025

mSystems

Response to reviewers' comments on submission mSystems01183-25

We thank the reviewers for their comments and feedback. Please find our responses to the issues raised below. The page and line numbers cited refer to the modified manuscript with changes tracked.

Sincerely,

Abderrahman Hachani (on behalf of all co-authors)

*Dr Abderrahman Hachani, PhD
Department of Microbiology and Immunology
Doherty Institute for Infection and Immunity
University of Melbourne
Parkville, Victoria, 3010
Ph. +61 414200062 | Email: ahachani@unimelb.edu.au*

Reviewer #1:

1. Some sections of the results and discussion are dense, with long sentences that may be difficult for readers less familiar with multi-omics.

Response: We have revised the results and discussion sections to improve readability. Modifications are highlighted in lines: 145-147, 150-155, 162-164, 378-380, 573-575, 754-791.

2. Additionally, the explanation of the DIABLO framework could be simplified or accompanied by an intuitive description to improve accessibility.

Response: We have expanded on the explanation of the DIABLO framework in the result section in lines 150-155: “DIABLO is a supervised multiblock method based on sparse Partial Least Squares Discriminant Analysis (sPLS-DA) (19, 20). DIABLO uses the PLS framework to model relationships between molecular features (derived from transcriptomic, proteomic or metabolomic datasets) and phenotypic outcomes. By integrating multiple omics datasets (defined as multiblock) and applying feature selection (sparsity), DIABLO highlights the subset of variables with the greatest discriminatory power – in this case, the staphylococcal molecules most strongly differentiating growth in RPMI versus serum (**Figure 1C-D, Figure S1A-C**).” We also include a schematic to support this explanation in Figure S1A.

3. Methodological details are generally clear, but some aspects could be expanded to ensure reproducibility—for example, serum preparation and fitness assays. It would also help to consistently indicate the number of biological replicates in the figure legends.

Response:

3.a) Serum preparation: The Material and Methods section has been modified to enhance reproducibility efforts by members of the community. The commercially sourced serum (Sigma Aldrich) used in our study was not heat-inactivated. We now also detail the processing of the serum in a new paragraph, “Serum treatment” of the Material and Methods (lines 883-892). An analysis of serum proteins by SDS-PAGE separation has also now been performed and results are shown in Figure S9. All phenotypic confirmation assays were performed with serum belonging to the same batch, except for phenotypic assays described in the new data presented in **Figure 3G** and **Figure S5E**, showing the increased production of staphyloxanthin by *S. aureus* WT JE2, upon exposure to serum. Considering reactive oxidative species are intrinsically labile, we confirmed our observations with 2 different serum batches.

3.b) Number of biological replicates in the figure legends: This has now been corrected, and the numbers of replicates are now indicated in all figure legends.

4. Simplify the DIABLO explanation.

Response: We have now simplified the explanation of the DIABLO framework in lines 150-155: “DIABLO is a supervised multiblock method based on sparse Partial Least Squares Discriminant Analysis (sPLS-DA) (19, 20). DIABLO uses the PLS framework to model relationships between molecular features (derived from transcriptomic, proteomic or metabolomic datasets) and phenotypic outcomes. By integrating multiple omics datasets (defined as multiblock) and applying

feature selection (sparsity), DIABLO highlights the subset of variables with the greatest discriminatory power – in this case, the staphylococcal molecules most strongly differentiating growth in RPMI versus serum (Figure 1C-D, Figure S1A-C)."

An illustration of the DIABLO process has also been included (Figure S1A).

5. Use of RPMI as only comparator. While defined and reproducible, it differs substantially from serum in protein content, nutrients and host factors such as heme, albumin and complement. Some observed bacterial responses may therefore reflect media differences rather than true host-specific adaptation.

Response: The rationale for using RPMI as the only comparator was established in our previous study (Mu *et al.*, Nat Communications, 2023. PMID: 36934086) where the selection for RPMI as comparative media to serum was described in the material and method section as “RPMI1640 was selected as the media as it produced characteristic growth curves for all four species and allowed standardisation of batched sample generation across the three omic platforms (i.e. transcriptomics, proteomics, and metabolomics).” RPMI is a defined medium and produces consistent results from experiments performed by research groups. Contrary to nutrient-rich media (Lysogeny Broth (LB), Tryptic-soy broth (TSB) or Brain Heart Infusion (BHI)) that promote rapid growth but could distort metabolic and regulatory states, RPMI attempts to mimic the nutrient-limitation of the host milieu encountered by *S. aureus* (Poudel *et al.*, 2020, PMID:32616573). Growth in RPMI is thus thought to provide more relevant gene expression profiles and linked phenotypes.

6. Consider whether heat-inactivated serum or additional controls could strengthen the physiological relevance.

Response: Besides using low speed centrifugation to clarify the serum from excess lipids that altered optical readings during fitness assays, this study used non-heated serum to prevent the loss of acellular immune elements such as immunoglobulins, complement, and host proteins that may alter the phenotypic readouts of NTML mutants in comparison with the WT strain. A paragraph describing the clarification process (Serum treatment, lines 883-892) was added in the material and method part of the manuscript, along a supplementary figure (Figure S9).

Using heat-inactivated serum has been considered to study the fitness of genes such as *vraX* involved in defence mechanisms, described in the literature as a complement evading staphylococcal element (see Figure 4, bottom left box and reference PMID: 28582645, and lines 754-756), but further investigation is beyond the scope of the current study.

7. Use of commercial serum: As noted by the authors, commercial serum may not fully recapitulate physiological serum composition, which could influence pathway activation and mutant phenotypes (e.g., *isdB*). This limitation should be emphasized in the discussion.

Response: We have now emphasized the differences between the serum used in the present study (commercially sourced) and the serum used to generate the omics dataset (Mu *et al.*, 2023) in the limitations section (lines 813-841).

8. Study *isdB* role under heme-containing conditions: Reassessing *isdB* mutant fitness in serum supplemented with hemoglobin or heme (physiological range), or in lysed RBC-

containing serum, would help explain the lack of phenotype in commercial serum and clarify the context-dependent utility of the *Isd* system.

Response: Our previous study (Mu et al., 2023, PMID: 36934086) used pooled sera from donors (Lifeblood, Melbourne, Australia). The processing of the pooled sera may have led to RBC lysis, as demonstrated by the reduced fitness of the *isdI* transposon mutant as compared to the WT Je2 strain (Supplementary Figure 10 from Mu et al. 2023) showing the growth kinetics of *S. aureus* WT JE2 vs the *tn::isdI* mutant in 50% serum, sourced from Lifeblood).

9. Quantify oxidative stress and antioxidant responses directly.

Measure intracellular ROS (e.g., H2DCFDA), catalase/peroxidase activities and staphyloxanthin levels (HPLC or spectrophotometry) in WT and *perR* mutants with and without serum or H₂O₂. This would provide functional support for the role of PerR and carotenoid pathways in oxidative protection beyond growth curves.

Response: We have addressed this suggestion by measuring the levels of staphyloxanthin, a carotenoid pigment produced by *S. aureus* to counteract oxidative stress. We now show in **Figure 3G** and **Figure S5E**, increased staphyloxanthin upon exposure to serum by JE2 wild type, as compared to the *perR::tn* and *crtM::tn* mutants. The material and method section has been updated with the paragraph “Quantification of Staphyloxanthin” (lines: 923-936).

10. Biofilm and adhesion assays on serum-coated surfaces

Evaluate whether serum exposure alters adherence or biofilm formation on host-like surfaces and test relevant mutants (e.g., *ecb*, *sbi*, *clfA*). This would link metabolic or vulnerability changes to colonization-relevant phenotypes.

Response: This would be an interesting experiment that we agree could yield new insights. However, given the large amount and complexity of the data we already present, we suggest these experiments would constitute a separate research project. We now mention the added value of such experiments in the discussion (line 762)

Minor Points

Line 121: "approximately": please indicate variance explained by PC1 for each dataset.

Response: this has been amended, lines 145-148.

Line 135: Pearson correlations >0.95 with only 5 isolates are surprisingly high. Consider adding p-values. Clarify whether replicates are treated as independent samples.

Response: In our analysis, the replicates represent biological replicates and were treated as independent samples in a within-subject (repeated-measures) design, as specified in the DIABLO model. The reported correlations (Pearson's $r > 0.95$) refer to the latent components extracted by the model rather than to correlations between individual features or the entire datasets. Because these latent components summarise the major shared sources of variation across omics layers, high correlations are expected and indicate that the model successfully maximised covariance according to the defined design matrix.

As relations are expected between transcriptomic, proteomic and metabolic layers, rather than hypothesis-testing, the DIABLO model achieved balanced covariance between datasets. Thus, statistical testing (e.g., p-values) of these correlation coefficients was not needed. We instead verified the model output using the *plotDIABLO()* function to ensure that component correlations were maximised across omics layers, as intended.

Figure 1E: "Proteome" should not be italicized; capitalize the first letter.

Response: this has been amended.

Serum composition: consider attempting to determine this more precisely.

Response: we have updated the materials and methods section, with the added paragraph "Serum treatment" associated with Figure S9.

Supplementary material: some plots are difficult to read (likely a PDF rendering issue).

Response: we have checked the images post-pdf rendering and have ensured these are legible following submission.

Line 216: missing period after "(Table S2 and S3)".

Response: this has been corrected.

Line 599: link contains a space and does not work.

Response: this has been corrected.

Reviewer #2 (Comments for the Author):

1. The authors mention these could be targets for future therapeutic development, but the manuscript does a poor job of providing details or examples of how this could be employed.

Response: Many pathogens rely on glycolysis and the tricarboxylic acid cycle for energy during bloodstream infection. We have referenced in the discussion (lines 670-671) a review on inhibitors of pyruvate kinase, succinate dehydrogenase, or isocitrate dehydrogenase that could impair bacterial energy production and virulence (Passalacqua, Charbonneau and O'Riordan, *Micro. Spectrum*, 2016, PMID: 27337445).

2. Metabolic interventions/inhibitors have not been highly successful in development (at least when used alone), so the authors could also state that in the limitation and/or discussion.

Response: We agree with this reviewer on the difficulty of developing bacterial metabolic inhibitors. We included itaconate as an example in the discussion (lines 670-672).

3. Finally, given the importance of the findings, it may be worthwhile to *reflect their findings on the burgeoning field of using alternative media* (e.g. mammalian cell culture growth) for susceptibility testing of antibiotics, which may be more akin to the serum environment with nutrient availability.

Response: we have addressed this in the limitations section (lines 841-845), where we suggest the use of Human Like Plasma Media (HPLM) that due to their composition, mimics more closely host-like conditions, for future studies instead of RPMI.

Re: mSystems01183-25R1 (**Integrated multi-omics reveals coordinated *Staphylococcus aureus* metabolic, iron transport and stress responses to human serum**)

Dear Dr. Abderrahman Hachani:

Your manuscript has been accepted, and I am forwarding it to the ASM production staff for publication. Your paper will first be checked to make sure all elements meet the technical requirements. ASM staff will contact you if anything needs to be revised before copyediting and production can begin. Otherwise, you will be notified when your proofs are ready to be viewed.

Sincerely,
Marnix Medema
Editor
mSystems

Reviewer #1 (Comments for the Author):

The authors have addressed all my comments and the revised version of the manuscript reads much better and has been greatly improved. I truly appreciate their effort.

Reviewer #2 (Comments for the Author):

The authors provide detailed responses to the prior critiques including additional experimental data to address concerns and improved limitation / discussion section.